# Associations of Dietary Riboflavin, Niacin, and Retinol with Age-related Hearing Loss: An Analysis of Korean National Health and Nutrition Examination Survey Data

**DOI:** 10.3390/nu11040896

**Published:** 2019-04-21

**Authors:** Tae Su Kim, Jong Woo Chung

**Affiliations:** 1Department of Otolaryngology, School of Medicine, Kangwon National University, 1 Gangwondaehakgil, Chuncheon, Gangwon-do 24341, Korea; kimtaesu77@gmail.com; 2Department of Otorhinolaryngology-Head & Neck Surgery, Asan Medical Center, University of Ulsan College of Medicine, 88 Olympic-ro 43 gil, Songpa-gu, Seoul 05505, Korea

**Keywords:** hearing loss, presbycusis, nutrients, riboflavin, niacin, retinol

## Abstract

Because age-related hearing loss (ARHL) is irreversible, prevention is very important. Thus, investigating modifying factors that help prevent ARHL is critical for the elderly. Nutritional status or nutritional factors for the elderly are known to be associated with many problems related to aging. Emerging studies suggest that there was the interaction between nutrition and ARHL. We aimed to investigate the possible impact of dietary nutrients on ARHL using data from the fifth Korean National Health and Nutrition Examination Survey (KNHANES) which included 4742 subjects aged ≥ 65 years from 2010 to 2012. All participants underwent an otologic examination, audiologic evaluation, and nutritional survey. The associations between ARHL and nutrient intake were analyzed using simple and multiple regression models with complex sampling adjusted for confounding factors, such as BMI, smoking status, alcohol consumption, and history of hypertension and diabetes. Higher intake groups of riboflavin, niacin and retinol was inversely associated with ARHL prevalence (riboflavin aOR, 0.71; 95% CI, 0.54–0.94; *p* = 0.016, niacin aOR, 0.72; 95% CI, 0.54–0.96; *p* = 0.025, retinol aOR 0.66; 95% CI, 0.51–0.86; *p* = 0.002, respectively). Our findings suggest the recommended intake levels of riboflavin, niacin, and retinol may help reduce ARHL in the elderly.

## 1. Introduction

Age-related hearing loss (ARHL) or presbycusis is a progressive, bilateral, symmetrical sensorineural hearing loss. ARHL is a result of degeneration of the cochlea or associated structures of the inner ear. Approximately ARHL affects 23% to 40% of the population older than 65 years of age [1]. 

Disabling hearing loss was defined as better ear hearing more than 40 dB (average of 0.5, 1, 2, and 4 kHz) in adults according to the World Health Organization (WHO) [2]. In 2012, the WHO estimated that there were 328 million adults with disabling hearing loss worldwide. Because of the aging population, the WHO suggested that more than 500 million will suffer from significant ARHL by 2025 [2]. ARHL is associated with increased cognitive dysfunction and dementia in the elderly [3,4,5], which can lead to social isolation, depression, and anxiety [6]. ARHL is irreversible; therefore, understanding potential modifying factors associated with ARHL is critical for prevention. 

Nutritional status or nutritional factors for the elderly are known to cause many problems related to aging, and emerging studies have suggested that there are interaction effects between nutrition and ARHL [7,8]. Metabolic syndrome is affected by nutritional status or nutritional factors [9], and we recently identified an association between metabolic syndrome and ARHL in a 5-year follow-up and large cohort study [10]. 

To reduce the risk of ARHL, it is important to identify protective nutritional factors. The Korean National Health and Nutrition Examination Survey (KNHANES) may be useful for determining whether or not there are protective nutritional factors. This survey collects comprehensive information, such as health status and nutritional condition from the general Korean population, using a complex, stratified, multistage, probability-clustered sampling method based on national census data.

Two studies have investigated the relationship between nutrition and hearing loss using KNHANES data. One study reported a relationship between vitamins and ARHL, but there was a limitation of research to the population included for one year only in 2011, rather than the total of 3 years from 2010 to 2012 in the fifth KNHANES [11]. The other study analyzed energy-related nutritional factors such as proteins, fats and carbohydrates, and found that low fat and low protein intake were associated with ARHL using the fifth KNHANES data, but defined hearing loss as ≥ 25 dB, which does not cause a major disability, rather than the ≥40 dB criterion for disabling hearing loss established by the WHO [12].

In this study, we aimed to investigate the possible impact of dietary nutrients on ARHL using data from the fifth KNHANES, which includes cross-sectional surveys of subjects from 2010 through 2012. The population of overall disabling hearing loss included 10% of total participants, and up to 40% in the population older than 65 years [13]. Because we wanted to analyze the relationship between ARHL and nutrition in the age group in which ARHL was mainly observed, we focused on this study only over 65 years of age.

## 2. Materials and Methods 

### 2.1. Study Participant

This study used data from the fifth KNHANES. The KNHANES is an ongoing nationwide epidemiological study conducted by the Korea Centers for Disease Control and Prevention of the Ministry of Health and Welfare. The institutional review board (IRB) of the Korean Centers for Disease Control and Prevention review and approve the KNHANES survey annually. All participants are randomly chosen from randomly assigned districts of cities and provinces in South Korea. Among the selected candidate population, 80.8% of individuals participated in the fifth KNHANES from 2010 to 2012. Written informed consent was obtained from all participants before commencement of the study.

During the period of the Fifth KNHANES, the total number of participants was 36,067 individuals. Among those 36,067 participants, 4742 subjects aged ≥ 65 years were included. Among the 4742 participants aged ≥ 65, 1022 participants were excluded because they had not received an audiometric test or had an abnormal tympanic membrane or had not completed the nutritional survey or had incomplete data (Figure 1). 

### 2.2. Otologic Examination and Audiologic Evaluation

During the fifth KNHANES survey, otologic examinations were conducted by ENT doctors by using a 4-mm, 0° endoscope (Xion GmbH, Berlin, Germany) to evaluate any abnormalities related to the tympanic membrane or middle ear including retraction, otitis media with effusion and cholesteatoma. Audiologic evaluation was performed by using a diagnostic audiometer (SA-203; Entomed, Malmoe, Sweden) at 0.5, 1, 2, 3, 4 and 6 kHz in a soundproof booth. Hearing impairment was defined as a hearing level exceeding an average of 40 dB on pure-tone audiometry at 0.5, 1, 2 and 4 kHz. Hearing impairment was divided into bilateral hearing loss, considered ARHL, and unilateral hearing loss. 

### 2.3. Evaluation of Nutrition Intake

Nutrition intake was surveyed using the complete 24-h recall method. All participants were instructed to continue their ordinary diets before the 24-h recall evaluation. Answers were not excluded by any certain days such as holidays or weekends. Nutrient intake was calculated by referencing the nutrient concentrations of foods described in the Korean Food Composition Table, which was devised by the Korean National Rural Resources Development Institute [14]. As shown in a previous report [15], the analyzed nutrient intake data included total energy (kJ/day (kcal/day)), protein (g/day), fat (g/day), carbohydrate (g/day), crude fiber (g/day), ash (g/day), calcium (mg/day), phosphorus (mg/day), iron (mg/day), sodium (mg/day), potassium (mg/day), β-carotene (µg/day), retinol (µg/day), thiamine (mg/day), riboflavin (mg/day), niacin (mg/day), and Vitamin C (mg/day). The intake of vitamin A (µg retinol equivalents (RE)/day) was calculated by addition of retinol (µg/day) and β-carotene/6 (µg/day). The subjects were divided into quartile groups of intake per day for each nutrient.

### 2.4. Statistical Analyses

The general characteristics between subjects with and without bilateral hearing impairment were compared using Student’s *t*-tests for continuous parameters and χ^2^ tests for categorical parameters. Nutrient intakes were analyzed by quartiles, and the lowest quartile was the reference group. Univariate analyses of each nutrient were performed to investigate factors associated with hearing impairment. Multivariate logistic regression analyses were used to confirm the association between hearing impairment and each nutritional factor. The analyses were adjusted for the following potential confounders: age, sex, body mass index (BMI), smoking status, alcohol consumption, hypertension, and diabetes mellitus. The results of the logistic regression analyses are expressed as odds ratios (OR) with 95% confidence intervals (CI). *p* values of <0.05 were considered statistically significant. All statistical analyses were performed using SPSS software (version 21.0; IBM Corp., Armonk, NY, USA).

## 3. Results 

Participant demographics are shown in Table 1. The occurrence of bilateral hearing loss exceeding an average of 40 dB was 17.88% (*n* = 665) in the over 65 years of age group. The average age of subjects with bilateral hearing loss was 75.4 ± 5.99 years old. Subjects with bilateral hearing impairment were older (*p* < 0.001), weighed less (*p* < 0.001), and were more likely to smoke (*p* = 0.0023) and to drink alcohol (*p* < 0.001) than those without bilateral hearing impairment.

Quartile values of each nutrient intake are shown in Table 2.

Table 3 shows the associations between nutrient intake and bilateral hearing impairment. The highest quartiles (75th–100th percentile) of all nutrients except for β-carotene showed significantly lower risk for bilateral hearing impairment (all *p* < 0.05) in the univariate analyses. In univariate analyses, the third quartiles (50th–75th percentile) of intake of fat (*p* = 0.007), protein (*p* = 0.005), ASH (*p* = 0.001), calcium (*p* = 0.007), iron (*p* = 0.011), retinol (*p* = 0.006), thiamin (*p* = 0.001), riboflavin (*p* = 0.006), niacin (*p* = 0.032), potassium (*p* = 0.005), and phosphorus (*p* = 0.001) showed significantly lower risk for bilateral hearing impairment. The second quartiles (25th–50th percentile) of carbohydrate (*p* = 0.024), fat (*p* = 0.037), calcium (*p* = 0.012), and iron (*p* = 0.040) showed significantly lower risk for bilateral hearing impairment in the univariate analyses. Multivariate analyses adjusted for age, sex, BMI, smoking status, alcohol consumption, hypertension, and diabetes mellitus showed significantly lower risk for bilateral hearing impairment in the highest quartiles of intake for riboflavin (aOR, 0.71; 95% CI, 0.54–0.94; *p* = 0.016), niacin (aOR, 0.72; 95% CI, 0.54–0.96; *p* = 0.025), and retinol (aOR 0.66; 95% CI, 0.51–0.86; *p* = 0.002).

Table 4 shows the results of the association between nutrient intake and unilateral hearing impairment. In univariate analyses, the third quartiles of intake of riboflavin (*p* = 0.042), retinol (*p* = 0.015), iron (*p* = 0.003), calcium (*p* = 0.022) and sodium (*p* = 0.003) showed significantly lower risk for unilateral hearing impairment. In addition, the highest quartiles of intake of β-carotene (*p* = 0.031), and vitamin A (*p* = 0.048) showed an inverse correlation with the prevalence of unilateral hearing loss. However, in multivariate analyses adjusted for age, sex, BMI, smoking status, alcohol consumption, hypertension, and diabetes mellitus showed no correlation between nutrient intake and unilateral hearing impairment. 

## 4. Discussion 

We found that dietary intake of riboflavin, niacin and retinol were associated with bilateral hearing impairment in elderly KNHANES participants, but there was no correlation between nutrient intake and unilateral hearing impairment. Overall, higher intake of riboflavin, niacin and retinol were related to a lower risk of ARHL.

According to the Dietary Reference Intakes for Koreans 2015, the average requirement of riboflavin is 1.0 mg/day and the recommended intake is 1.2 mg/day for ages 65 years and older [16]. In this study, only the highest quartile of riboflavin intake satisfied the average requirement and the recommended intake. The average requirement of niacin was 11 mg/day, and the recommended intake is 14 mg/day for age 65 years or older. The average requirement of niacin was met in the third and fourth quartiles of niacin intake, but only the highest quartile met the recommended intake. Retinol is defined in the dietary reference intakes for Koreans. 

Increased serum level or dietary intake of retinol decreased the incidence of ARHL in various countries such as Asian and Western countries [17,18,19]. Higher riboflavin intakes in western countries are significantly associated with better hearing thresholds [18], but there have been no reports in Asian countries. In a Korean study, dietary intake of niacin was inversely correlated with ARHL, but there was no significant difference after adjusting for age, sex, smoking history, BMI, and medical history (diabetes, hypertension) [11]. Vitamin C was reported to be associated with better hearing in a Korean study [11], but there was no relationship between intake of vitamin C and ARHL in some studies [18,20]. In addition, our results also showed that vitamin C was not associated with ARHL. 

There have been several studies evaluating the relationship between retinol and hearing, as retinol has a high concentration in the inner ear [21], and retinoic acid, an active metabolite of retinol, contributes to development of the organ of Corti [22]. The relationship between retinol and hearing loss was reported in many of these studies [8,17,18,19,21,22,23,24]. Higher intake of retinol was associated with better hearing in women [19]. In regarding to the dietary intake of retinol, the risk of ARHL (>40 dB HL) was lower in the highest quintile compared to the lowest quintile in older adults [18]. Increased serum retinol has also been associated with reduced incidence of ARHL [17]. The role of retinol in the prevention of hearing loss has not yet been determined. Reported potential mechanisms for the prevention of hearing loss suggest that retinoic acid inhibits an apoptosis-related c-Jun N-terminal kinase signal pathway in the inner ear [23], or that it facilitate regeneration of auditory hair cells [24]. Another possible way that retinol may prevent hearing loss lies in its antioxidant properties [25], although there is some controversy about whether retinol is an antioxidant [26]. 

Riboflavin also has an antioxidant role, especially with respect to lipid peroxidation and reperfusion oxidative injury [27]. Riboflavin, by itself or as a part of glutathione reductase, is a fundamental component of the antioxidant mechanisms within cell systems. Riboflavin affects the process of glutathione conversion and shows an antioxidant activity. By glutathione reductase (GR) with riboflavin (in the flavin adenine dinucleotide coenzyme form), oxidized glutathione is converted to its reduced form (GSH) [28]. GSH serves as an endogenous antioxidant in every cell in the body [29]. Riboflavin can decrease oxidative damages as free radical scavengers [25]. It also reduces reactive oxygen species (ROS) and peroxides such as hydroperoxides. Therefore, riboflavin deficiency may lead to increased risk of ARHL due to oxidative stress and ROS production. 

ARHL is caused by irreversible loss of sensory hair cells and/or degeneration of spiral ganglion neurons. Niacin increases synaptic plasticity and axonal growth in a rat model of stroke [30]. Such findings suggest that niacin-induced neuroprotective effects increase brain-derived neurotrophic factor (BDNF)/tropomyosinreceptor kinase B (TrKB) pathways after a stroke [30]. The survival of spiral ganglion neurons (SGNs) of the cochlear is retained by endogenous neurotrophic support [31]. Therefore, the risk of ARHL may be reduced by upregulated BDNF by niacin intake. When the dietary niacin intake was higher, vascular endothelial cell function increased [32]. Endothelial dysfunction could lead to interruption of vascular flow of the inner ear. In idiopathic sudden sensorineural hearing loss (ISSHL), compromised vascular endothelial cell function has been reported [33]. Therefore, improved vascular endothelial cell function induced by niacin could reduce the risk for ARHL. 

This report may be the first community-based epidemiological study to reveal that higher intake of niacin is inversely associated with the prevalence of ARHL in the elderly. However, we could not find the causal mechanism of the higher intake of niacin and ARHL. Therefore, measurement of serum of niacin in ARHL patients or animal study with niacin deficiency model will be needed to understand the relationship between niacin and hearing.

In multivariate analysis, higher dietary intake of riboflavin, niacin and retinol had a strong correlation with ARHL. These results were fairly consistent with previous studies in which a healthy diet including multivitamins influenced better hearing [34,35]. Vitamin intake decreased with age [11]; therefore, supplementary vitamin intake may be more effective for hearing in the elderly.

Retinol and riboflavin are mainly found in foods of animal origin like milk, cheese, and eggs. Niacin is found in a variety of whole and processed foods, including meat from various animal sources, seafood, and spices. The principal aspects of traditional Korean diet include proportionally high consumption of rice and vegetables, moderate consumption of fish, and low consumption of red meat and dairy products [36]. Over the last few decades, the Korean diet has changed from traditional food, mainly composed of rice and vegetables, to a westernized diet, rich in red meat and dairy products, in line with development and globalization [37]. However, the elderly in Korea may not have enough riboflavin, niacin, and retinol, because they maintain traditional Korean diets, including low consumption of meat and milk.

As a cross-sectional epidemiological survey, our study cannot establish causal relationships between the nutritional intake and ARHL. To identify causal relationships, an experimental study using human or animal would be needed. Furthermore, nutritional intake was estimated based on the 24-h recall method of the subjects. Although trained staff interviewed the participants, responses were dependent on the subjects’ memory, leaving open the possibility of measurement error. Serum analysis of nutrient levels could be useful when investigating their direct association with ARHL, because nutritional bioavailability may vary from person to person. 

## 5. Conclusions 

In conclusion, our study found that higher intake of riboflavin, niacin and retinol were inversely correlated with prevalence of bilateral hearing loss in the ≥65 years age group. ARHL was lower in the groups taking niacin and riboflavin above the recommended intake. Although Koreans have continued to adopt more western diets, traditional grain- and vegetable-oriented eating habits are still maintained by the elderly. Therefore, encouraging the consumption of dairy products and meat to ensure sufficient supplies of riboflavin, niacin, and retinol in older populations may help reduce the incidence of ARHL. 

## Figures and Tables

**Figure 1 nutrients-11-00896-f001:**
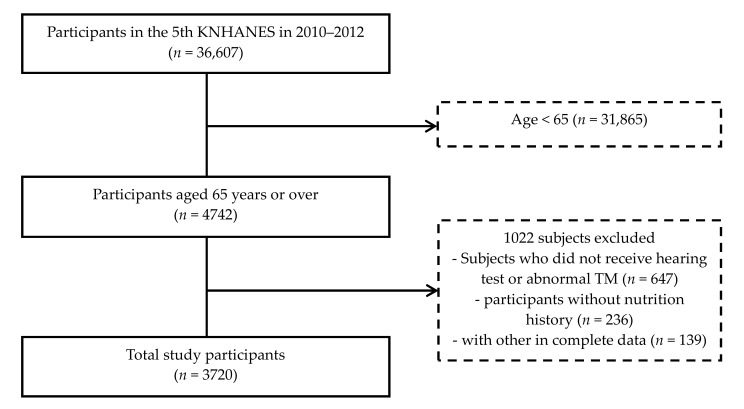
A schematic illustration of participant selection in the present study. TM, tympanic membrane.

**Table 1 nutrients-11-00896-t001:** Baseline characteristics of study participants.

Characteristics	Bilateral Age-Related Hearing Loss (≥40 dB)		*p* Value
	No (*n* = 3055)	Yes (*n* = 665)	
**Age**	71.78 ± 4.83	75.40 ± 5.99	<0.001
Gender			<0.001
Male	1298 (42.5)	337 (50.7)	
Female	1757 (57.5)	328 (49.3)	
BMI (kg/m²)			<0.001
Underweight (<18.5)	106 (3.5)	41 (6.2)	
Normal (≥18.5, <23)	1873 (61.3)	458 (68.9)	
Over (≥23)	1076 (35.2)	166 (24.9)	
Smoking			0.0023
Never	1869 (61.2)	364 (54.7)	
Former	843 (27.6)	200 (30.1)	
Current	343 (11.2)	101 (15.2)	
Alcohol consumption			<0.001
Non	1523 (49.9)	387 (58.2)	
in month<1	482 (15.8)	70 (10.5)	
1≤in month≤4	535 (17.5)	103 (15.5)	
2≤in week	515 (16.8)	105 (15.8)	
Diabetes			0.87
No	2475 (81.0)	537 (80.8)	
Yes	580 (19.0)	128 (19.2)	
Hypertension			0.197
No	1441 (47.2)	332 (49.9)	
Yes	1614 (52.8)	333 (50.1)	

Data are presented as mean ± SD, number (%). BMI, body mass index.

**Table 2 nutrients-11-00896-t002:** Quartile categories of nutrient intake.

Nutrient	Quartile 1	Quartile 2	Quartile 3	Quartile 4
Total energy intake (kcal)	<1239.64	1239.64–1599.27	1599.28–2028.59	2028.60≤
Carbohydrate (g/day)	<229.03	229.03–296.03	296.04–367.13	367.14≤
Fat (g/day)	<10.16	10.16–17.89	17.90–31.13	31.14≤
Protein (g/day)	<35.03	35.03–48.96	48.97–68.07	68.08≤
Fiber (g/day)	<3.81	3.81–5.76	5.77–8.69	8.70≤
Ash (g/day)	<9.74	9.74–14.54	14.55–21.07	21.08≤
Calcium (mg/day)	<217.14	217.14–350.01	350.02–543.39	543.40≤
Iron (mg/day)	<6.85	6.85–10.58	10.59–16.71	16.72≤
Potassium (mg/day)	<1601.38	1601.38–2289.28	2289.29–3224.92	3224.93≤
Sodium (mg/day)	<2104.36	2104.36–3376.17	3376.18–5195.31	5195.32≤
Phosphorus (mg/day)	<683.91	683.91–920.32	920.33–1217.36	1217.37≤
Retinol (μg/day)	<3.22	3.22–15.51	15.52–60.55	60.56≤
β-carotene (μg/day)	<1020.92	1020.92–2237.26	2237.27–4412.40	4412.41≤
Vitamin A (μg Retinol Equivalents /day)	<208.76	208.76–426.34	426.35–816.56	816.57≤
Thiamin (mg/day)	<0.64	0.64–0.90	0.91–1.29	1.30≤
Riboflavin (mg/day)	<0.50	0.50–0.74	0.75–1.13	1.14≤
Niacin (mg/day)	<8.26	8.26–11.59	11.60–16.35	16.36≤
Vitamin C (mg/day)	<36.17	36.17–66.36	66.37–114.93	114.94≤

**Table 3 nutrients-11-00896-t003:** Odds ratios of nutritional factors and bilateral hearing loss.

Nutrient	Univariable Analysis		Multivariable Analysis		Nutrient	Univariable Analysis		Multivariable Analysis	
	OR (95% CI)	*p*	OR (95% CI)	*p*		OR (95% CI)	*p*	OR (95% CI)	*p*
**Total Energy**					Retinol				
Q1	Reference		Reference		Q1	Reference		Reference	
Q2	0.77(0.61–0.98)	0.030 *	0.82(0.63–1.06)	0.132	Q2	0.97(0.78–1.22)	0.819	1.05(0.82–1.34)	0.700
Q3	0.80(0.63–1.00)	0.054	0.87(0.67–1.13)	0.285	Q3	0.72(0.59–0.91)	0.006 *	0.79(0.61–1.02)	0.075
Q4	0.66(0.52–0.84)	0.001 *	0.80(0.61–1.06)	0.125	Q4	0.61(0.48–0.78)	0.001 *	0.66(0.51–0.86)	0.002 *
Carbohydrate					β-carotene				
Q1	Reference		Reference		Q1	Reference		Reference	
Q2	0.76(0.60–0.97)	0.024 *	0.88(0.68–1.14)	0.321	Q2	1.10(0.88–1.39)	0.412	1.23(0.96–1.58)	0.105
Q3	0.86(0.68–1.08)	0.198	0.92(0.71–1.19)	0.527	Q3	0.90(0.71–1.14)	0.364	1.08(0.83–1.39)	0.576
Q4	0.73(0.58–0.93)	0.010 *	0.93(0.71–1.22)	0.588	Q4	0.80(0.63–1.01)	0.065	1.00(0.77–1.31)	0.981
Fat					Vitamin A				
Q1	Reference		Reference		Q1	Reference		Reference	
Q2	0.78(0.62–0.99)	0.037 *	0.92(0.72–1.18)	0.522	Q2	0.95(0.76–1.20)	0.683	1.03(0.80–1.32)	0.828
Q3	0.73(0.58–0.92)	0.007 *	0.81(0.62–1.04)	0.102	Q3	0.83(0.65–1.04)	0.108	0.99(0.77–1.28)	0.915
Q4	0.62(0.49–0.79)	0.001 *	0.76(0.58–1.01)	0.052	Q4	0.72(0.56–0.91)	0.006 *	0.86(0.66–1.12)	0.270
Protein					Thiamin				
Q1	Reference		Reference		Q1	Reference		Reference	
Q2	0.86(0.68–1.08)	0.185	0.99(0.77–1.27)	0.919	Q2	0.87(0.70–1.09)	0.228	0.98(0.76–1.25)	0.845
Q3	0.71(0.56–0.90)	0.005 *	0.87(0.67–1.14)	0.312	Q3	0.66(0.53–0.84)	0.001 *	0.78(0.60–1.02)	0.070
Q4	0.63(0.49–0.80)	0.001 *	0.75(0.57–1.01)	0.053	Q4	0.64(0.51–0.82)	0.001 *	0.82(0.62–1.08)	0.152
Fiber					Riboflavin				
Q1	Reference		Reference		Q1	Reference		Reference	
Q2	0.85(0.68–1.07)	0.175	.97(0.75–1.25)	0.798	Q2	0.82(0.65–1.03)	0.085	0.91(0.71–1.17)	0.459
Q3	0.91(0.72–1.14)	0.413	1.11(0.86–1.43)	0.418	Q3	0.72(0.58–0.91)	0.006 *	0.86(0.67–1.11)	0.258
Q4	0.67(0.52–0.85)	0.001 *	.89(0.68–1.17)	0.404	Q4	0.56(0.44–0.71)	0.001 *	0.71(0.54–0.94)	0.016 *
ASH					Niacin				
Q1	Reference		Reference		Q1	Reference		Reference	
Q2	0.80(0.64–1.00)	0.051	0.87(0.68–1.12)	0.278	Q2	1.03(0.83–1.30)	0.775	1.19(0.93–1.42)	0.174
Q3	0.64(0.61–0.81)	0.001 *	0.76(0.58–0.01)	0.054	Q3	0.77(0.61–0.98)	0.032 *	0.95(0.73–1.23)	0.675
Q4	0.62(0.49–0.79)	0.001 *	0.77(0.59–1.01)	0.062	Q4	0.59(0.46–0.76)	0.001 *	0.72(0.54–0.96)	0.025 *
Calcium					Vitamin C				
Q1	Reference		Reference		Q1	Reference		Reference	
Q2	0.74(0.59–0.94)	0.012 *	0.91(0.70–1.17)	0.439	Q2	0.83(0.66–1.05)	0.115	0.97(0.75–1.24)	0.790
Q3	0.73(0.58–0.92)	0.007 *	0.95(0.73–1.23)	0.679	Q3	0.81(0.65–1.02)	0.079	0.99(0.77–1.28)	0.945
Q4	0.68(0.54–0.86)	0.001 *	0.88(0.67–1.14)	0.327	Q4	0.63(0.50–0.80)	0.001 *	0.83(0.64–1.09)	0.180
Iron					Potassium				
Q1	Reference		Reference		Q1	Reference		Reference	
Q2	0.79(0.62–0.99)	0.040 *	0.88(0.68–1.13)	0.314	Q2	0.88(0.70–1.10)	0.252	0.99(0.78–1.28)	0.978
Q3	0.74(0.59–0.93)	0.011 *	0.89(0.68–1.15)	0.357	Q3	0.72(0.57–0.90)	0.005 *	0.91(0.70–1.17)	0.454
Q4	0.71(0.56–0.89)	0.004 *	0.88(0.68–1.14)	0.330	Q4	0.57(0.45–0.72)	0.001 *	0.79(0.60–1.04)	0.091
Sodium					Phosphorus				
Q1	Reference		Reference		Q1	Reference		Reference	
Q2	0.88(0.70–1.10)	0.263	0.96(0.75–1.24)	0.754	Q2	0.97(0.78–1.21)	0.776	1.12(0.88–1.43)	0.365
Q3	0.86(0.69–1.09)	0.216	0.96(0.74–1.24)	0.730	Q3	0.67(0.53–0.85)	0.001 *	0.77(0.59–1.01)	0.055
Q4	0.71(0.56–091)	0.004 *	0.85(0.65–1.12)	0.251	Q4	0.59(0.46–0.75)	0.001 *	0.76(0.57–1.01)	0.055

Data are presented OR (95% CI). Statistics were carried out using Logistic regression. Multivariable analysis was adjusted for age, sex, body mass index (BMI), smoking status, alcohol consumption, hypertension, and diabetes mellitus. CI, confidence interval; OR, odds ratio. * *p* < 0.05.

**Table 4 nutrients-11-00896-t004:** Odds ratios of nutritional factors for unilateral hearing loss.

Nutrient	Univariable Analysis		Multivariable Analysis		Nutrient	Univariable Analysis		Multivariable Analysis	
	OR (95% CI)	*p*	OR (95% CI)	*p*		OR (95% CI)	*p*	OR (95% CI)	*p*
**Total Energy**					Retinol				
Q1	Reference		Reference		Q1	Reference		Reference	
Q2	1.07(0.84–1.35)	0.587	1.03(0.81–1.32)	0.783	Q2	0.96(0.77–1.21)	0.726	0.96(0.76–1.21)	0.741
Q3	1.03(0.81–1.31)	0.808	1.02(0.79–1.31)	0.893	Q3	0.74(0.59–0.95)	0.015 *	0.83(0.57–1.03)	0.054
Q4	1.06(0.84–1.34)	0.629	1.02(0.78–1.33)	0.895	Q4	0.86(0.68–1.09)	0.213	0.88(0.69–1.11)	0.281
Carbohydrate					β-carotene				
Q1	Reference		Reference		Q1	Reference		Reference	
Q2	1.21(0.95–1.54)	0.116	1.20(0.94–1.54)	0.145	Q2	0.87(0.69–1.10)	0.240	0.86(0.68–1.09)	0.212
Q3	1.13(0.89–1.43)	0.327	1.12(0.87–1.44)	0.364	Q3	0.83(0.66–1.04)	0.110	0.85(0.67–1.07)	0.166
Q4	1.15(0.91–1.46)	0.246	1.15(0.89–1.49)	0.284	Q4	0.77(0.61–0.98)	0.031 *	0.79(0.62–1.01)	0.059
Fat					Vitamin A				
Q1	Reference		Reference		Q1	Reference		Reference	
Q2	0.96(0.76–1.21)	0.721	0.92(0.73–1.17)	0.510	Q2	0.91(0.73–1.15)	0.446	0.90(0.71–1.14)	0.378
Q3	0.88(0.69–1.11)	0.278	0.85(0.67–1.09)	0.206	Q3	0.81(0.64–1.03)	0.084	0.83(0.65–1.06)	0.134
Q4	0.92(0.73–1.17)	0.511	0.90(0.70–1.16)	0.432	Q4	0.79(0.62–0.99)	0.048 *	0.80(0.63–1.03)	0.080
Protein					Thiamin				
Q1	Reference		Reference		Q1	Reference		Reference	
Q2	0.85(0.67–1.08)	0.185	0.82(0.64–1.05)	0.119	Q2	1.19(0.94–1.50)	0.152	1.20(0.94–1.53)	0.143
Q3	0.95(0.76–1.20)	0.678	0.93(0.73–1.18)	0.535	Q3	1.05(0.83–1.34)	0.670	1.05(0.82–1.35)	0.693
Q4	0.91(0.72–1.14)	0.403	0.88(0.68–1.13)	0.310	Q4	1.02(0.80–1.29)	0.902	1.01(0.78–1.31)	0.936
Fiber					Riboflavin				
Q1	Reference		Reference		Q1	Reference		Reference	
Q2	1.02(0.81–1.29)	0.860	0.97(0.76–1.23)	0.783	Q2	0.88(0.70–1.11)	0.289	0.88(0.70–1.12)	0.309
Q3	0.85(0.67–1.07)	0.164	0.84(0.66–1.08)	0.179	Q3	0.78(0.62–0.99)	0.042 *	0.78(0.61–1.01)	0.057
Q4	0.87(0.69–1.10)	0.253	1.04(0.81–1.33)	0.783	Q4	0.85(0.67–1.07)	0.173	0.85(0.66–1.09)	0.189
ASH					Niacin				
Q1	Reference		Reference		Q1	Reference		Reference	
Q2	0.96(0.76–1.21)	0.720	0.97(0.76–1.23)	0.783	Q2	0.96(0.76–1.21)	0.718	0.92(0.72–1.18)	0.505
Q3	0.84(0.66–1.07)	0.162	0.84(0.66–1.08)	0.179	Q3	1.01(0.80–1.28)	0.905	1.03(0.80–1.31)	0.839
Q4	1.01(0.80–1.28)	0.906	1.04(0.81–1.33)	0.783	Q4	0.95(0.75–1.20)	0.673	0.92(0.71–1.19)	0.510
Calcium					Vitamin C				
Q1	Reference		Reference		Q1	Reference		Reference	
Q2	0.93(0.74–1.17)	0.518	0.91(0.72–1.16)	0.438	Q2	0.83(0.66–1.04)	0.110	0.82(0.65–1.05)	0.109
Q3	0.76(0.60–0.96)	0.022 *	0.75(0.58–1.03)	0.054	Q3	0.85(0.67–1.07)	0.157	0.84(0.66–1.07)	0.148
Q4	0.88(0.70–1.11)	0.287	0.89(0.69–1.13)	0.334	Q4	0.80(0.63–1.01)	0.057	0.81(0.63–1.03)	0.087
Iron					Potassium				
Q1	Reference		Reference		Q1	Reference		Reference	
Q2	0.78(0.62–0.98)	0.034 *	0.86(0.60–1.04)	0.065	Q2	0.87(0.69–1.10)	0.257	0.92(0.72–1.16)	0.470
Q3	0.70(0.56–0.89)	0.003 *	0.80(0.55–1.03)	0.056	Q3	0.89(0.70–1.12)	0.312	0.88(0.69–1.13)	0.304
Q4	0.84(0.67–1.05)	0.130	0.83(0.65–1.06)	0.135	Q4	0.87(0.69–1.10)	0.257	0.89(0.69–1.15)	0.364
Sodium					Phosphorus				
Q1	Reference		Reference		Q1	Reference		Reference	
Q2	0.78(0.62–0.98)	0.034 *	1.02(0.80–1.30)	0.903	Q2	0.91(0.72–1.15)	0.435	0.90(0.71–1.15)	0.398
Q3	0.70(0.56–0.89)	0.003 *	0.99(0.77–1.27)	0.931	Q3	0.98(0.78–1.24)	0.859	0.97(0.77–1.24)	0.787
Q4	0.84(0.67–1.05)	0.130	1.07(0.83–1.38)	0.587	Q4	0.90(0.71–1.14)	0.367	0.88(0.68–1.14)	0.320

Data are presented OR (95% CI). Statistics were carried out using Logistic regression. Multivariable analysis was adjusted for age, sex, body mass index (BMI), smoking status, alcohol consumption, hypertension, and diabetes mellitus. CI, confidence interval; OR, odds ratio. * *p* < 0.05.

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
