# Peer review of "Associations of Dietary Riboflavin, Niacin, and Retinol with Age-related Hearing Loss: An Analysis of Korean National Health and Nutrition Examination Survey Data"

_nutrients, 2019, doi:10.3390/nu11040896_

Reviewer 1 Report

ARHL is an important impairement to the well-being of elderly populations, and all efforts aimed to prevent it are worth trying. The authors address this problem by identifying protective nutritional factors by analyzing a cohort of Korean people > 65 years old. Results obtained from a complete nutritional evaluation were confronted with data from otologic revisions. Multivariate analysis highlighted a protective effect of a high consumption of riboflavin, niacin and retinol.

The manuscript is well written and the main points are clearly exposed. Furthermore, it addresses an important point that is mostly unresearched.

My main criticism is that the results section is very short. I would recommend to expand it and to discuss more in deep even the negative results (these are also important), comparing them with similar works done on other populations. This is important to evaluate whether the results here presented are of general interest or they describe a particular of the Korean population. I also miss some information on the genetics of ARHL, genes involved, SNPs etc. ¿Do the authors have access to genetic data on the cohort studied?  

Lastly, and as a general reader, I also miss some information on the  traditional Korean diet, (i.e. what are the sources for riboflavin, niacin and retinol?), and how are these changing with the westernalization of the diet.

Author Response

Dear reviewer,

Thank you very much for your comments. Below is the authors' response. Also, we revised the manuscript and the changes are highlighted.

----------------------------------------------------------------------

My main criticism is that the results section is very short.

I would recommend to expand it and to discuss more in deep even the negative results (these are 

also important), comparing them with similar works done on other populations.

This is important to evaluate whether the results here presented are of general interest or they describe a particular of the Korean population.

Thank you for your kind advice. We further described negative findings in the result section and discussed more comparing with similar studies on other populations. Thus, we explained the difference between our study and other studies in the discussion section.

I also miss some information on the genetics of ARHL, genes involved, SNPs etc. ? Do the authors have access to genetic data on the cohort studied? 

Genetic data were not collected in this study. We agree that genetic research with large cohort is needed, considering there are many genetic effects on hearing loss. Nevertheless, our research is of high value, because our study used extensive and well-organized data.

Lastly, and as a general reader, I also miss some information on the traditional Korean diet, (i.e. what are the sources for riboflavin, niacin and retinol?), and how are these changing with the westernalization of the diet.

Thank you for your kind advice. We describe the information of the traditional Korean diet and sources for riboflavin, niacin, and retinol. And we mentioned the necessity for meat and dairy products in the discussion section.

Best wishes,

Jong Woo Chung

Reviewer 2 Report

The study is described well and the conclusions are statistically validated. In their methods section, the authors did not mention noise exposure or ototoxic drug exposure as factors that would exclude participants. Did the authors consider these potential risk factors for hearing loss in the study population?

Author Response

Dear reviewer,

Thank you very much for your comments. Below is the authors' response. Also, we revised the manuscript and the changes are highlighted.

-----------------------------------------------------------------------

The study is described well and the conclusions are statistically validated.

In their methods section, the authors did not mention noise exposure or ototoxic drug exposure as factors that would exclude participants.

Did the authors consider these potential risk factors for hearing loss in the study population?

Thank you for your kind advice. But there was no survey of the ototoxic drug exposure in the Korean National Health and Nutrition Examination Survey. Although there was a questionnaire about noise exposure, there was no questionnaire on whether the hearing loss was experienced after noise exposure. Thus we couldn’t exclude these potential risk factors in our study.
Best wishes,

Jong Woo Chung
